# Learning of Discrete Graphical Models with Neural Networks

**Abhijith J.**
Centre for High Energy Physics,
Indian Institute of Science, Bengaluru.
abhijithj@iisc.ac.in

**Andrey Y. Lokhov, Sidhant Misra, Marc Vuffray**
Theoretical Division,
Los Alamos National Laboratory
{ lokhov, sidhant, vuffray }@lanl.gov

## Abstract

Graphical models are widely used in science to represent joint probability distributions with an underlying conditional dependence structure. The inverse problem of learning a discrete graphical model given i.i.d samples from its joint distribution can be solved with near-optimal sample complexity using a convex optimization method known as Generalized Regularized Interaction Screening Estimator (GRISE). But the computational cost of GRISE becomes prohibitive when the energy function of the true graphical model has higher order terms. We introduce NeurISE, a neural net based algorithm for graphical model learning, to tackle this limitation of GRISE. We use neural nets as function approximators in an Interaction Screening objective function. The optimization of this objective then produces a neural-net representation for the conditionals of the graphical model. NeurISE algorithm is seen to be a better alternative to GRISE when the energy function of the true model has a high order with a high degree of symmetry. In these cases NeurISE is able to find the correct parsimonious representation for the conditionals without being fed any prior information about the true model. NeurISE can also be used to learn the underlying structure of the true model with some simple modifications to its training procedure. In addition, we also show a variant of NeurISE that can be used to learn a neural net representation for the full energy function of the true model.

## 1 Introduction

Joint probability distributions of random variables with underlying conditional dependence structure are ubiquitous in science and engineering. The dependency structure of these distributions often reflect the properties of the scientific models that generate them. Undirected graphical models, also known as Markov random fields, are a natural way to represent such distributions and have found use in myriad fields such as physics [5], artificial intelligence [31], and biology [23] to name a few.

Due to the importance of graphical models, the inverse problem of learning them given i.i.d samples from their joint distribution has been a very active area of research. This problem was first addressed by Chow and Liu [8], who solved it for the case of graphical models with a tree structure. Since then there have been many works on learning graphical models under certain assumptions about the true model, with most of them focused on the special case of learning Ising models [28, 18, 26]. Bresler [4] gave the first polynomial time greedy algorithm that learned Ising models without any underlying assumptions. But the number of samples required to learn the model (i.e. sample complexity) using this method was still sub-optimal. The first near-sample optimal method to learn binary models with second order interactions (Ising models) was introduced by Vuffray *et al.* [29]. Under this approach, the learning problem is converted to a convex optimization problem which reconstructs the neighborhood of each variable in the model. The generalization of this method to learn any discrete graphical model is known as the *Generalized Regularized Interaction Screening Estimator*

(GRISE) [30]. This algorithm runs in polynomial time in the size of the model and its sample complexity is close to known information-theoretic lower bounds [27].

Despite being near sample optimal, the computational cost of GRISE becomes very high when trying to learn models with higher order interactions. To learn a model that has interactions up to order $L$, GRISE will need to have all possible models at that order in its hypothesis space. For a $p$ variable model with $L$-order interactions the computation complexity of GRISE goes as $\tilde{O}(p^L)$, which can be high for large $L$. Even if the true model has a high degree of symmetry or structure that reduces the effective number of parameters to be learned, GRISE will not be able to leverage this in a significant way to reduce the size of its hypothesis space.

In this work we propose a way to overcome this shortcoming arising from the linear parameterization that GRISE uses to represent candidate models. We introduce a method, which we call *Neural Interaction Screening Estimator* (NeurISE), that elegantly combines the strength of GRISE and the non-linear representation power of neural networks. As neural nets are universal function approximators [1, 17], NeurISE has the ability to learn the true model given enough samples just like GRISE. In addition, we demonstrate experimentally that NeurISE is much more efficient than GRISE for learning higher order models with some form of underlying symmetry. We exhibit practical examples where this performance gain can be exponentially large with a parameter space reduction of 99.4% already on small systems of only 7 variables. Another important aspect of graphical model learning is learning the structure of the conditional independence relations between random variables. While one may think at first that the use of a neural network could obfuscate the graphical model structure, we show that with a proper regularization process, the Markov random field structure reappears within the weights of the neural network. Finally, we also provide an aggregated cost function for learning a global probability distribution or energy function using consensus of local NeurISE reconstructions.

Throughout this work we will compare NeurISE to GRISE as it is the current state of the art method for learning undirected discrete graphical models, both theoretically from a sample complexity point of view [30] and empirically [22] when compared to other methods. Other existing recent methods for learning undirected graphical models include greedy [16] and pseudo-likelihood [20, 22, 32] type approaches. On the directed graphical models side, one can mention Directed Acyclic Graph (DAG) based methods that are popular for structure learning in the continuous variable setting [21]. Such methods are not directly related to this work as we are specifically interested in learning the discrete graphical model from which our samples are drawn that is undirected and may have an arbitrary underlying graph structure. Finally, other related line of work deals with testing in graphical models [10, 2] and estimations from single samples [7, 3, 11, 15, 9, 12].

The paper is organized as follows. The interaction screening principle is explained in Section 2 and NeurISE is presented in Section 3. Structure learning with NeurISE using an appropriate initialization and $\ell_1$ regularization is illustrated in Section 4. The method for representing the energy function of the true model with a single neural net representation for the energy function is discussed in Section 5. The supplementary material contains additional experimental results with NeurISE.

## 2 Learning graphical models via interaction screening

### 2.1 The interaction screening method

We consider a graphical model defined over $p$ discrete variables $\sigma_i \in [q]$ for $i \in [p]$, where the notation $[k]$ refers to the set containing exactly $k \in \mathbb{N}$ elements. The models we consider will be positive probability distributions over the set of $p$ dimensional vectors, $\underline{\sigma} \in [q]^p$. Without loss of generality this probability distribution can be written as,

$$\mu(\underline{\sigma}) = \frac{1}{Z} \exp\left(H(\underline{\sigma})\right). \tag{1}$$

The function, $H : [q]^p \to \mathbb{R}$, is called the *energy function* or the *Hamiltonian* of the graphical model. The quantity $Z$ is called the partition function and it ensures normalization. GRISE considers a linear parameterization of $H(\underline{\sigma})$ by expanding it with respect to a chosen basis

$$H(\underline{\sigma}) = \sum_{k \in \mathcal{K}} \theta_k^* g_k(\underline{\sigma}_k), \tag{2}$$

where $g_k$ denote the elements of the basis, $\mathcal{K}$ denotes the index set of the basis functions acting on the variables $\underline{\sigma}_k \subseteq \underline{\sigma}$, and $\theta_k^*$ are the parameters of the model. Given $n$ i.i.d. samples $\underline{\sigma}^{(1)}, \ldots, \underline{\sigma}^{(n)}$, GRISE uses the convex *interaction screening objective* (ISO) to estimate the parameters $\underline{\theta}^*$ around one variable at a time. For any $u \in [p]$, the ISO is given by

$$\textbf{ISO:} \quad \mathcal{S}_n(\underline{\theta}_u) = \frac{1}{n} \sum_{t=1}^{n} \exp\left( - \sum_{k \in \mathcal{K}_u} \theta_k g_k(\underline{\sigma}_k^{(t)}) \right), \tag{3}$$

where $\mathcal{K}_u = \{ k \in \mathcal{K} \mid \sigma_u \in \underline{\sigma}_k \}$ and $\underline{\theta}_u$ are the parameters associated with $\mathcal{K}_u$. The quantity

$$H_u(\underline{\sigma}) = \sum_{k \in \mathcal{K}_u} \theta_k g_k(\underline{\sigma}_k) \tag{4}$$

denotes the *partial energy function* containing all terms dependent on $\sigma_u$ and is directly related to the conditional distribution $\mu(\sigma_u \mid \underline{\sigma}_{\setminus u})$. In the presence of prior information in the form of an $\ell_1$-bound on the parameters, the estimation can be efficiently performed by:

$$\textbf{GRISE:} \quad \min_{\underline{\theta}_u} \mathcal{S}_n(\underline{\theta}_u), \quad \text{subject to:} \ \|\underline{\theta}_u\|_1 \leq \gamma. \tag{5}$$

The basis functions in (5) are assumed to be centered: $\sum_{\sigma_u} g_k(\underline{\sigma}_k) = 0$ for all $k \in \mathcal{K}_u$. A quick intuition behind the minimization in (5) can be obtained by considering the ISO in the limit $n \to \infty$

$$\lim_{n \to \infty} \mathcal{S}_n(\underline{\theta}_u) \to \mathcal{S}(\underline{\theta}_u) = \mathbb{E}\left[ \exp\left( - \sum_{k \in \mathcal{K}_u} \theta_k g_k(\underline{\sigma}_k) \right) \right]. \tag{6}$$

It is easy to verify using simple computation that $\nabla_{\theta_k} \mathcal{S}(\underline{\theta}_u^*) = 0$. Since $\mathcal{S}$ is a convex function, the minimization in (5) estimates the parameters correctly in the limit of infinite samples. We refer the reader to [30] for a detailed finite sample analysis of GRISE.

## 2.2 Basis function hierarchies

The choice of basis functions in (3) plays a crucial role in the computational complexity of GRISE. Unless clearly specified by the application, one must use generic complete hierarchies of basis functions that have the ability to express any discrete function $H(\underline{\sigma})$. We present two such generic heirarchies below.

**Centered indicator basis:** This basis is defined by using the one-dimensional centered indicator functions given by

$$\Phi_s(\sigma) \colon = \begin{cases} 1 - \frac{1}{q}, & \text{if } s = \sigma, \\ -\frac{1}{q}, & \text{otherwise.} \end{cases} \tag{7}$$

For any $\mathcal{K} \subset 2^{[p]}$ a set of basis functions can be constructed as

$$\Phi_{\underline{s}_k}(\underline{\sigma}_k) = \prod_{i \in k} \Phi_{s_i}(\sigma_i) \quad \text{for each} \quad k \in \mathcal{K}, \ \underline{s}_k \in [q]^{|k|}. \tag{8}$$

**Monomial basis:** For the special case of binary variables with $\sigma_i \in \{-1, 1\}$, we can define the monomial basis for any $\mathcal{K} \subset 2^{[p]}$ as

$$g_k(\underline{\sigma}_k) = \prod_{i \in k} \sigma_i \quad \text{for each} \quad k \in \mathcal{K}. \tag{9}$$

When $\mathcal{K} = 2^{[p]}$ both the *centered indicator basis* and the *monomial basis* are complete. However, this choice makes GRISE, as given in (5), clearly intractable. Without any known strong underlying structure, a natural, and perhaps the only logical choice, is to restrict the so-called *interaction order* to a specified value $L \leq p$ by considering $\mathcal{K} = \{ k \in 2^{[p]} \mid |k| \leq L \}$. The complexity of GRISE is driven by the number of terms in the exponent in the objective which is now bounded by $O(p^L)$. A natural hierarchy of basis functions is constructed by starting from $L = 1$, and increasing in small steps as required. We thus obtain increasing representation power at the expense of higher computational cost. This approach is highly effective when the interaction order of the true underlying model is low. However, for models with high interaction order this approach can be computationally expensive, even if the model has significant structure.

# 3 NeurISE: Neural Interaction Screening Estimator

To deal with higher order models more easily, we use GRISE with a neural net ansatz to represent the partial energy function. A neural net, being a universal function approximator, will eventually cover the space of models as its size is increased and provides a natural alternative to the monomial and centered indicator hierarchies in Section 2.2. But it explores the function space in a different way, and given the ability of neural nets to find patterns in data, it is to be expected that this approach will work better in practice for learning structured models with high interaction order.

## 3.1 Neural net parameterization of partial energy function

The analysis of GRISE in [30] relies on the linearity of $H(\underline{\sigma})$ in the parameters and the centered property of $g_k$, none of which is true for a neural net parameterization. Nevertheless, our approach using neural nets attempts to generalize the intuition regarding the zero gradient property of the infinite-sample limit of ISO in (6). Similar to GRISE, we propose a neural network parameterization for one variable $u$ at a time given by approximating the partial energy function $H_u$ in (4) as

$$H_u(\underline{\sigma}) \approx \tilde{H}_u(\underline{\sigma}, w) = \langle \Phi(\sigma_u), \mathrm{NN}_u(\underline{\sigma}_{\backslash u}, w) \rangle = \sum_{s=1}^{q} \Phi_s(\sigma_u) \mathrm{NN}_u(\underline{\sigma}_{\backslash u}, w)(s), \qquad (10)$$

where $\Phi(\sigma_u) = \{\Phi_1(\sigma_u), \ldots, \Phi_q(\sigma_u)\}$ and the function $\mathrm{NN}_u(\underline{\sigma}_{\backslash u}, w)$ in (10) is a vector valued function with $q$ outputs. The input to the neural net $(\underline{\sigma}_{\backslash u})$ is the set of all variables expect $\sigma_u$. We use $w$ to denote the weights of the neural net and they serve the role of the parameters $\underline{\theta}$ in (4). The representation above is automatically centered in $\sigma_u$. Moreover, the representation does not lose any generality since the global energy function can always be written as

$$H(\underline{\sigma}) = H_{\backslash u}(\underline{\sigma}_{\backslash u}) + H_u(\underline{\sigma}) = H_{\backslash u}(\underline{\sigma}_{\backslash u}) + \sum_{a=1}^{q} \Phi_a(\sigma_u) \, H_{u,a}(\underline{\sigma}_{\backslash u}). \qquad (11)$$

The corresponding neural net interaction screening objective (NeurISO) is given by

$$\textbf{NeurISO:} \quad L_u(w) = \frac{1}{n} \sum_{t=1}^{n} \exp\left(-\langle \Phi(\sigma_u^{(t)}), \, \mathrm{NN}(\underline{\sigma}_{\backslash u}^{(t)}; w) \rangle\right), \quad \text{for each } u \in [p]. \qquad (12)$$

**An intuitive justification for NeurISO – a variational argument:** Consider the traditional GRISE with a complete set of the *centered indicator basis* defined in (8) with all terms, i.e., order $L = p$. Due to completeness of the basis, optimizing over the parameters $\underline{\theta}$ is equivalent to optimizing over the set of all discrete functions $f : [q]^p \to \mathbb{R}$ that are centered w.r.t. $\sigma_u$. Since GRISE is able to recover the correct energy function using this basis, it follows that the true partial energy function $H_u$ is a *global optimum* of the following variational problem using the infinite-sample ISO:

$$H_u(\underline{\sigma}) = \operatorname{argmin}_{f:[q]^p \to \mathbb{R}} \mathbb{E}\left[\exp(-f(\underline{\sigma}))\right] \quad \text{subject to:} \quad \sum_{\sigma_u} f(\underline{\sigma}) = 0. \qquad (13)$$

Since the objective in (13) is convex in $f$, the partial energy function $H_u$ is the *global optimum* of the problem. When the size of a neural net is sufficiently large, minimizing the NeurISO in (12) is almost equivalent to the variational problem in (13). Although the function $\tilde{H}_u$ is a non-convex function of $w$, a property similar to the zero gradient property of $\mathcal{S}(\underline{\theta}_u)$ can be shown to hold for $L_u$ in the limit of infinite samples and neural net size when the function space covered by the neural net is large enough such that there exists a set of weights $w$ such that $H_u = \tilde{H}_u$. In this setting, we can show that one of the minima of $L_u$ corresponds to $H_u$. The gradient of $L_u$ w.r.t. $w$ can be written as

$$\frac{\partial L_u}{\partial w_j} = -\frac{1}{Z} \sum_{\underline{\sigma}} \langle \Phi(\sigma_u), \, \frac{\partial \, \mathrm{NN}(\underline{\sigma}_{\backslash u}; w)}{\partial w_j} \rangle \exp\left(H_u(\underline{\sigma}) - \langle \Phi(\sigma_u), \mathrm{NN}(\underline{\sigma}_{\backslash u}; w) \rangle + H_{\backslash u}(\underline{\sigma}_{\backslash u})\right).$$

If $H_u(\underline{\sigma}) = \tilde{H}_u(\underline{\sigma}, w)$, for all values of $\underline{\sigma}$, the gradient is zero. We will call the minima for which this condition is satisfied as the *interaction screening minima*.

Since neural networks are universal function approximators [1], we see from the variational problem in (13) that the interaction screening minima are the global optima of $L_u$ over $w$ in the limit of infinite

number of samples. This important observation shows that NeurISE always has the capability to learn a neural network representation for the true graphical model from which the samples came from. Even for finite size, this observation motivates the use of SGD or its variants to optimize $L_u$. The inherent noise in SGD will prevent it from being stuck in any spurious local minima and it will converge more easily to the interaction screening minima. It is further accompanied by its usual perks of parallelizability and the ability to implement using GPUs.

NeurISE as described so far, doesn't learn the full energy function. Instead it gives $p$ neural nets which approximate the partial energy function of each variable in the model. This gives us an approximation for the conditionals of the true model. Let $\text{NN}_u^*$ be the fully trained neural net obtained by minimizing $L_u$. For the true model the conditional probability of a variable conditioned on everything else can be estimated as

$$\mu[\sigma_u|\underline{\sigma}_{\backslash u}] = \frac{\exp(\sum_{a=1}^q \Phi_a(\sigma_u),\ H_{u,a}(\underline{\sigma}_{\backslash u}))}{\sum_{s=1}^q \exp(\sum_{a=1}^q \Phi_a(s),\ H_{u,a}(\underline{\sigma}_{\backslash u})))} \approx \hat{\mu}[\sigma_u|\underline{\sigma}_{\backslash u}] = \frac{\exp(\langle \Phi(\sigma_u),\ \text{NN}_u^*(\underline{\sigma}_{\backslash u})\rangle)}{\sum_{s=1}^q \exp(\langle \Phi(s),\ \text{NN}_u^*(\underline{\sigma}_{\backslash u})\rangle)}.$$
(14)

The conditionals can be used to draw samples from the learned model using Gibbs sampling [14].

**Remark:** Everything in the above discussion carries over to the case of binary models by using

$$H(\underline{\sigma}) = H_{\backslash u}(\underline{\sigma}_{\backslash u}) + \sigma_u H_u(\underline{\sigma}_{\backslash u}) \approx H_{\backslash u}(\underline{\sigma}_{\backslash u}) + \tilde{H}_u(\underline{\sigma}_u) = H_{\backslash u}(\underline{\sigma}_{\backslash u}) + \sigma_u \text{NN}(\underline{\sigma}_{\backslash u}; w), \quad (15)$$

where $\text{NN}(; w)$ is a scalar valued function.

### 3.2 Experiments

Now we will test NeurISE on two highly structured graphical models. In our testing we will compare NeurISE to GRISE. These are completely different types of algorithms and finding the right metric to compare them is tricky. GRISE converts the learning problem into a convex optimization problem for which theoretical guarantees can be derived. On the other hand, NeurISE is a non-convex problem but the learning process in this case can be easily parallelized on a GPU using off-the-shelf machine learning libraries. If there are no limitations on the computational power, then both these methods will find the true model eventually. But on real hardware the performance of these will depend on implementation and on the true model being learned. To quantify the hardness of these algorithms in a device independent fashion, we compare the number of free parameters these algorithms optimize over per variable ($N_p$). Roughly, this number reflects the dimension of the hypothesis space of these algorithms.

We will be using feed forward neural nets with the swish activation function (swish($x$) = $x \cdot$ sigmoid($x$)) [25] . We specify the size of a neural net with two numbers, $d$ and $w$, which will be the number of hidden layers in the model and the number of neurons in each hidden layer respectively. All the nets were trained using the ADAM optimizer [19].

#### 3.2.1 Learning binary models with higher order interactions

We expect NeurISE to work well on models with a high degree of underlying structure even if that model has higher order interactions. GRISE, without any prior information about the structure, will need to use the entire hierarchy described in Section 2.2 to recover the correct model.

To demonstrate this, we generate samples from a graphical model with the following energy function,

$$H(\underline{\sigma}) = \sum_{l=1}^L \theta_l^* \sum_{i=1}^{p-l+1} \sigma_i \ldots \sigma_{i+l-1}, \quad (16)$$

and learn it using GRISE and NeurISE [1]. The hypergraph structure of this model is one dimensional and it has up to $L$ order interactions. We impose an extra symmetry here by choosing the same interaction strength for all terms of the same order. So in effect there are only $L$ parameters to be learned here. But for our experiments the learning algorithms will be unaware of this property and also of the one dimensional nature of the model. First we compute the $\ell_1$ error in the learned

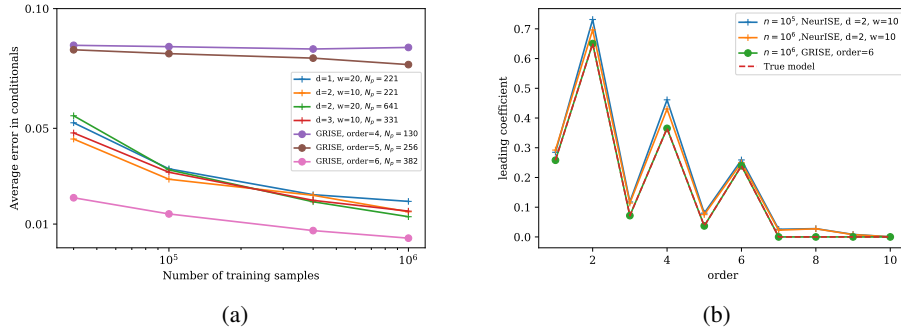

(a)                                                                  (b)

Figure 1: Learning the model given by Eq. (16), with $p = 10$ and $L = 6$. (a) $\ell_1$ error in the learned conditionals averaged over all possible inputs. (b) The absolute value of the leading coefficient of the learned model at each order compared to that of the true model.

conditionals averaged over all possible inputs using Eq. (14). We compare the average error in the learned conditionals for a $p = 10$ variable model with $L = 6$ in Fig. 1a. The $\theta$ values are chosen from $[-1, 1]$.

We also study our learned model by expanding the learned neural nets in the monomial basis. This can be done for any binary function using standard formulae [24]. If the neural net learns the correct model, the leading coefficients in the monomial expansion at each order should match those of the true model at that order. The absolute values of these coefficients are compared in Fig.1b. Both these metrics are exponentially expensive in $p$ to compute. The small model makes these explicit comparisons possible without resorting to Gibbs sampling.

We see from Fig. 1a that GRISE with all terms up to fifth order ($L = 5$), just one order lower than the true model, fails to learn the model correctly. For this algorithm the number of input samples has little effect on the error in the conditionals. This implies that the candidate models considered by the algorithm are far away from the true model in function space. Including all terms up to fifth order in GRISE requires us to optimize over 256 free parameters. A neural net model comparable to this is the $[d = 2, w = 10]$ model which has 221 free parameters to optimize. We see that this model has better error in comparison to GRISE with similar $N_p$. Also the error in this case decays as the number of training samples increases, unlike the floor observed in L=5-GRISE. This is an indication that the neural net manages to learn the true model.

Looking at the other neural net models, we see that most of them learn the true model well. The best algorithm according to Fig. 1a is GRISE with sixth order interactions included. Ths is expected because it has the advantage of having the true model in its hypothesis space. On the other hand, with NeurISE we can only get close to the true model. But the advantage of NeurISE is that it can do this with far fewer parameters than L=6-GRISE with 382 parameters.

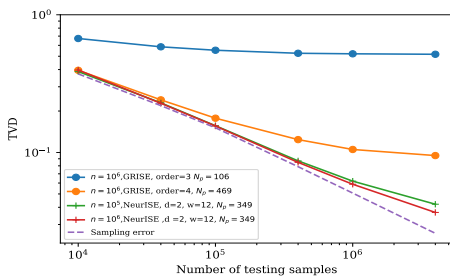

Figure 2: Error in TVD in the learned models when the true model is a 15 variable, sixth order model as given in Eq. (16). The GRISE order is chosen so that it has approximately the same number of free parameters as NeurISE. The x-axis gives the number of samples drawn from the model after learning it.

In Fig. 1b, we see that the neural net learns the coefficients well up to sixth order. But it cannot completely suppress the higher order coefficients. This happens because NeurISE implicitly has higher order polynomials in its hypothesis space. The agreement with the true model improves as the number of training samples increases and the spurious hypotheses are suppressed.

The efficiency of NeurISE over GRISE is obvious for models with a higher number of variables. For larger models we have to resort to sampling from the models and computing the total variation distance (TVD) between the sampled distributions. To set a base line for sampling error we take two independent set of samples from the true model and compare the TVD between them. If the learned model is close to the true model then the TVD between their samples should closely follow this base line. GRISE with sixth order param-

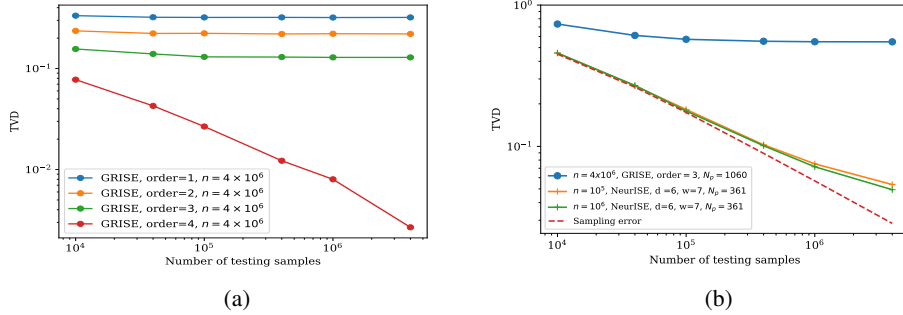

(a)                                          (b)

Figure 3: Learning the GHZ state (a) GRISE on the 4 qubit state showing the presence of terms up to the highest order (b) Total variation distance between the distributions sampled from the GHZ state on 7 qubits and those sampled from the learned models.

eters for a 15 variable model is a 3472 variable optimization problem. This was intractable on the hardware used for these experiments. Instead we compare GRISE with up to fourth order interactions ($N_p = 469$) with [$d = 2, w = 12$] NeurISE ($N_p = 349$). This comparison is given in Fig. 2. We see that NeurISE learns the true model well with fewer number of parameters in comparison to GRISE with a higher $N_p$, and performs better than GRISE even with fewer training samples.

### 3.2.2   Learning a $q = 4$ model with permutation symmetry

Now we test NeurISE on graphical models over $\{1, 2, 3, 4\}^p$. Additionally these distributions will also have complete permutation symmetry, i.e. the probability of a string will not change under permutations of that string,

$$\mu(\underline{\sigma}_1, \underline{\sigma}_2, \ldots, \underline{\sigma}_p) = \mu(\underline{\sigma}_{\pi(1)}, \underline{\sigma}_{\pi(2)}, \ldots, \underline{\sigma}_{\pi(p)}), \ \ \forall \, \pi \in S_p, \tag{17}$$

where $S_p$ is the symmetric group on $p$ elements. Distributions with this symmetry occur naturally in quantum physics. For this specific experiment we will learn the probability distribution obtained by measuring a quantum state called the *GHZ state on $p$ qubits*. We will work with the distribution obtained from the GHZ state by measuring it in a basis known as the *tetrahedral POVM*. The mathematical details of this setup can be found in Ref. [6]. Measuring a $p$ qubit GHZ state in this basis produces a positive distribution on $[4]^p$ which is symmetric under permutations. This means that this distribution can be represented as a Gibbs distribution. Our testing on small systems show the energy function of this distribution has all terms up to order $p$. (Fig. 3a). Fig. 3b shows the error measured in TVD when learning a GHZ state on 7 qubits. Doing full GRISE in this case is prohibitively expensive ($N_p = 62500$). But NeurISE performs remarkably well with a small neural net ($N_p = 361$).

## 4   Structure learning with input regularization

In this section we will discuss structure learning with NeurISE, where the focus is to learn the structure of the hypergraph associated with the true model. The ensuing discussion will focus on binary models for simplicity, but the same principle applies for all alphabets.

If NeurISE converges to an interaction screening minima then $NN_u^*$ will contain information about the sites in the neighbourhood of $u$ in the hypergraph of the true model. More precisely, for inputs in $\{1, -1\}^{p-1}$, the output of $NN_u^*$ will be insensitive to the inputs corresponding to sites outside the neighbourhood of $u$. By looking at which inputs influence the output, we can learn the underlying dependency structure of the graphical model. If a certain input doesn't influence the output, then we expect the input weights connecting that input to the rest of the net to be close to zero. But, $NN_u^*$ is a function whose full domain is $\mathbb{R}^{p-1}$ which we are restricting to $\{1, -1\}^{p-1}$. This restriction is a many to one map in the space of functions, i.e. there are many functions with a continuous domain that can be projected to the same function with a discrete domain. This means that $NN_u^*$ could be a function that depends on all its inputs when its domain is continuous. It could very well have non-zero weights at certain inputs while being not sensitive to those inputs when they take discrete values.

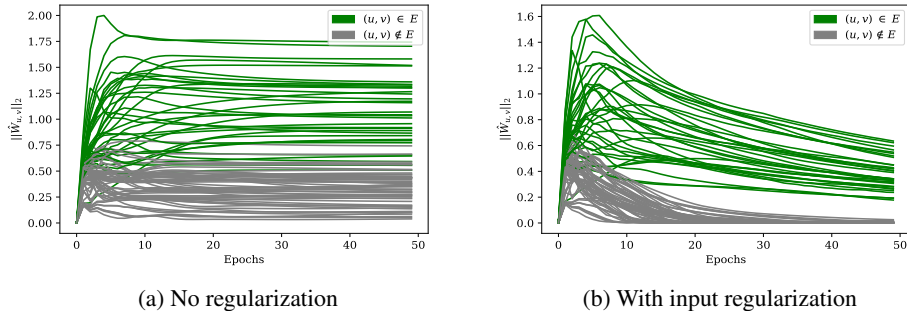

(a) No regularization          (b) With input regularization

Figure 4: The effect of regularization on the training of input weights. [$p = 10$, $\alpha = 0.2$, $\beta = 1.2$, d=2, w = 10, $n = 4 \times 10^5$].

This problem can be fixed in practice by taking two steps. First, at the beginning of training the input weights must be initialized to zero. Secondly, we must regularize the NeurISE loss function with the $\ell_1$ norm of only the input weights. Both these steps will ensure that the weights corresponding to the inputs that do not influence the output go to zero.

We test this method on pairwise binary models on random graphs. The energy function here will be $H(\underline{\sigma}) = \sum_{(u,v) \in E} \theta^*_{u,v} \sigma_u \sigma_v$. The random graph is generated by the Erdős-Rényi model [13] and the interaction strengths are chosen uniformly random from an interval $[\alpha, \beta]$. We denote by $\hat{W}_{u,v}$ the array of input weights of $\mathrm{NN}^*_u$ that connect the input corresponding to site $v$ to the rest of the network. In Fig. 4, we see the effect regularization and initialization has on $||\hat{W}_{u,v}||_2$ while NeurISE is being trained. Regularization forces the non-edge weights to zero and clearly separates them from the edge weights. This means that by plotting a histogram of $||\hat{W}_{u,v}||_2$ values at the end of training, we can distinguish between the edges and non edges in the graph. This is demonstrated in Fig. 5, where a 20 variable random graph is reconstructed perfectly from the histogram of trained weights. For a model with higher order interactions, NeurISE will be able to learn neighborhood of each variable. This information can then even be used as a prior in GRISE to reduce its computational cost.

In the case of GRISE there is an exact formula that lets us choose a value for the regularization penalty parameter. The theoretical arguments used to derive that formula do not apply to NeurISE. Yet, the $O(\sqrt{\ln(p)/n})$ formula used for GRISE (Theorem 1, [29]) is seen to be a good rule of thumb for choosing the regularization penalty for NeurISE as well. A low penalty will produce a flat histogram of input weights with no cluster near zero. While a high penalty would force all the input weights to be close to zero. If there are enough samples available for structure learning, then the correct regularization penalty must set the non-edge input weights to zero while keeping the edge input weights at non-zero. Nevertheless, Fig. 4a shows that even if the penalty is too low the input weight histogram will order the edges correctly i.e, the non-edges will have lower weights than the edges. So even in this case there exists a threshold that can separate the edges from the non-edges. The correct regularization penalty makes this threshold more evident in the histogram. As the edges are ordered correctly we can always choose a low threshold if we want to avoid true edges being classified as non-edges by the algorithm. And vice-versa, we can choose a high threshold if we want to avoid non-edges in the true model being classified as edges by the algorithm. More experiments on structure learning and further discussions on choosing this threshold can be found in the supplementary material.

## 5 Learning the complete energy function with NeurISE

NeurISE as described so far learns the conditionals of the graphical model. At the level of the energy function, the learned model for a particular variable represents the partial energy function of that variable. But for some applications it would be beneficial to have the complete energy function.

Reconstructing the energy function from the partial energies is a non-trivial task. If all the partial energies are compatible, i.e. if they come from the same underlying energy function, then we can

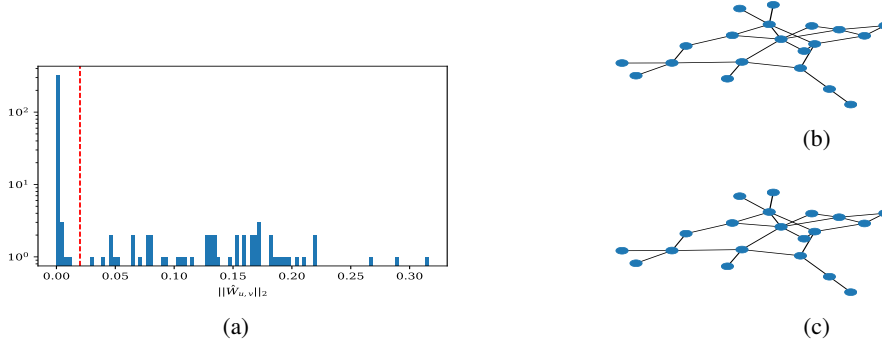

(a)                                                    (c)

Figure 5: Structure learning on a random graph of average degree of 2.6 [$p = 20$, $\alpha = 0.3$, $\beta = 1.3$, d=2, w = 10, $n = 4 \times 10^5$] (a) The histogram of $||\hat{W}_{u,v}||_2$ after training. The vertical red line is the threshold used to distinguish edges from non edges. (b) Graph of the true model. (c) Graph reconstructed from the histogram.

expand them in the monomial basis and reconstruct the energy function from the expansion. But this method is computationally expensive. Instead a simple modification to the NeurISE loss function can let us learn the complete energy function directly. We will explain this for the case of binary models, but a similar principle can be used for models with general alphabets as well. The modification to learn the energy function is based on (15) which shows that the partial energy for a variable $u$ can be written as,

$$H_u(\underline{\sigma}) = \sigma_u H_u(\underline{\sigma}_{\setminus u}) = \frac{1}{2}\left(H(\underline{\sigma}) - H(\underline{\sigma}_{\sim u})\right), \tag{18}$$

where $\underline{\sigma}_{\sim u}$ is $\underline{\sigma}$ with the variable $u$ flipped in sign. Using a neural net as a candidate for the energy function $H(\underline{\sigma}) \approx \text{NN}(\underline{\sigma}; w)$, we can rewrite the loss in Eq. (3) as

$$L_u(w) = \frac{1}{n}\sum_{t=1}^{n}\exp\left(\frac{\text{NN}(\underline{\sigma}_{\sim u}; w) - \text{NN}(\underline{\sigma}; w)}{2}\right) \tag{19}$$

To ensure that the trained neural net gives the correct energy function we have to sum up these individual loss functions to construct a single loss function,

$$L(w) = \sum_{u=1}^{p} L_u(w) = \frac{1}{n}\sum_{u=1}^{p}\sum_{t=1}^{n}\exp\left(\frac{\text{NN}(\underline{\sigma}_{\sim u}; w) - \text{NN}(\underline{\sigma}; w)}{2}\right). \tag{20}$$

Just as before, we can show that if the neural net has sufficient expressive power and if $n \to \infty$, then the global minima of this loss function correspond to the correct energy function. For GRISE this modification is not necessary as it learns directly in the monomial basis. The results of learning a model with this loss function is given in the supplementary material due to space considerations.

## Broader impact.

We believe that this work, as presented here, has no direct ethical impact or societal consequences. But, our work paves way for learning higher order graphical models on real world data sets. There are many unanswered questions about NeurISE that are relevant to such real-world applications. For instance, can any theoretical guarantees be given on the structure learned by NeurISE? Or, can we modify NeurISE to learn a model free of certain biases present in the training data set? We hope to answer some of these questions in the future.

## Acknowldegments

We acknowledge support from the Laboratory Directed Research and Development program of Los Alamos National Laboratory under project numbers 20190059DR, 20190195ER, 20190351ER, and 20210078DR.

## Footnotes

[1]The code and data for this can be found at `https://github.com/lanl-ansi/NeurISE`.

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
