[Supplementary Material]

# Learning of Discrete Graphical Models with Neural Networks
# Supplementary Material

**Abhijith J.**
Centre for High Energy Physics,
Indian Institute of Science, Bengaluru.
abhijithj@iisc.ac.in

**Andrey Y. Lokhov, Sidhant Misra, Marc Vuffray**
Theoretical Division,
Los Alamos National Laboratory
{ lokhov, sidhant, vuffray }@lanl.gov

This document contains supplementary materials for the paper "Learning of Discrete Graphical Models with Neural Networks". Here we show results of some more experiments done with NeurISE. Section A has results of learning the Ising model. Section B has more results on structure learning, including learning hypergraphs. Section C has results on learning the full energy function using NeurISE.

## A   Learning Ising models.

For this experiment we learn Ising models with two body interactions. We take random graphs with an average degree of three and choose the interaction strengths uniformly at random from [-1,1]. The Hamiltonian here has the from,

$$H(\underline{\sigma}) = \sum_{i<j} \theta_{ij}^* \sigma_i \sigma_j. \tag{1}$$

This is an adversarial experiment for NeurISE when compared to GRISE. GRISE will learn this model in the second level of its hierarchy with $O(p)$ parameters per optimization. The neural net will have to fit a linear function of its inputs, which it will not be able to do as well as low-degree GRISE. Despite this, NeurISE does a good job of learning the true model, albeit with more number of free parameters when compared to second order GRISE.

Figure 1: Learning a random Ising model (a) $\ell_1$ error in the learned conditionals averaged over all possible inputs for a 10 variable model (b) TVD between samples drawn from the learned models and those drawn from the true model for a 15 variable model. The neural net used here is [d=3, w=15].

# B  Structure learning with NeurISE.

## B.1  Learning hypergraphs

We show that NeurISE can accurately reconstruct the neighbourhood of each variable for a general model with higher order interactions. In Fig 2 we learn the following 15 variable model [1],

$$H(\underline{\sigma}) = \frac{1}{2}\sigma_1\sigma_3\sigma_5\sigma_7\sigma_9 + \theta^*_{1,15}\ \sigma_1\sigma_{15} + \sum_{i=1}^{14}\theta^*_{i,i+1}\ \sigma_i\sigma_{i+1} \qquad (2)$$

The $\theta^*$ parameters here are chosen uniformly from $[0.3, 1.3]$. As seen in Fig. 2b, NeurISE can perfectly reconstruct the neighbourhoods of each variables. The fifth order term shows up as clique of size 5 connecting the corresponding variables. Once the neighborhood reconstruction is done, we can use this as a prior in GRISE. This can reduce the number of free parameters in $L-$order GRISE from $O(p^L)$ to $O(D^L)$, where $D$ is the size of the neighbourhood of the variable being learned.

(a)

(b)

Figure 2: Structure learning on the energy function in Eq. (2) [$p = 15$, $\alpha = 0.3$, $\beta = 1.3$, d=2, w = 15, $n = 10^6$]. (a) The histogram of $||\hat{W}_{u,v}||_2$ after training. The vertical red line is the threshold used to distinguish edges from non edges. (b) Reconstructed graph. The neighbourhood of every variable is learned perfectly

## B.2  Learning graphs when the number of samples is too low

The success of structure learning depends on the number of samples used in the algorithm. The number of samples required to perfectly learn the structure depends on the strength of interaction and the degree of the underlying model. This is reflected in the sample complexity lower bound which is exponential in the product of the degree of the graph and the maximum strength of interaction . In particular, learning models with higher degree with a limited number of samples makes distinguishing edges from non-edges more difficult. The histogram of trained inputs weights in this case will be more spread out as seen in Fig. 3a. Despite this there are only a few mis-classified edges in the reconstructed graph. In the histogram these edges usually lie close to the large cluster of weights close to zero. If the threshold line is chosen right after this large cluster most edges and non-edges are classified accurately. GRISE also exhibits a similar behaviour when the number of samples available are inadequate for perfect structure learning .

## B.3  Accuracy of structure learning

Here we will look at the accuracy of structure learning with NeurISE over randomized experiments.

The results of structure learning for various classes of models is plotted in Fig.4. Each data point in these plots is a result of 20 randomized experiments. The thresholds for structure learning where chosen automatically by constructing the distribution of input weights and using an outlier detection

(a)

(b)

(c)

Figure 3: Structure learning on a random graph of average degree of 3.6 [$p = 20$, $\alpha = 0.3$, $\beta = 1.3$, d=2, w = 10, $n = 10^6$] (a) The histogram of $||\hat{W}_{u,v}||_2$ after training. The vertical red line is the threshold used to distinguish edges from non edges. (b) Graph of the true model. (c) Graph recor                                    e

(a) Random second order model with 30 variables, average degree= 3, $d = 2$, $w = 10$.

(b) 20 variable model in Eq.(2) with the fifth order term randomized, average degree = 3, $d = 2$, $w = 20$.

(c) Random third order model with 30 variables, average degree = 2, $d = 2$, $w = 10$

(d) Random fourth order model with 30 variables, average degree = 2, $d = 2$, $w = 10$

Figure 4: Accuracy of structure learning over randomized experiments with confidence intervals. For every model we take, $[\alpha, \beta] = [0.3, 1.3]$. Every data point is result of 20 randomized experiments. The "Total Accuracy" here refers to the accuracy of the algorithm for learning both edges and non-edges in the graph. The average degree of each hypergraph is the average number of neighbours of each node in the hypergraph

method to isolate the weights clustered around zero. The edges that were a fraction of the standard deviation away from the mean of the distribution were labelled as edges. The value of this fraction is a fixed value for each of the four experiments in Fig.4. This is fixed first by running a few test experiments and inspecting the histograms of their input weights. The other hyper parameters for learning are also fixed in this fashion. In general, visual inspection of the histogram gives a better value for the threshold. But this is impractical if one has to run many randomized experiments. The method based on the standard deviation of the histogram automates the structure learning process.

Systematically, these experiments show the difference of the neural network hierarchy from the polynomial hierarchy. The sample complexity of learning a neural net representation of a random fourth order model is much smaller than that of learning a random second order model or a random third order. The neural net can learn a random fourth order order with higher accuracy consistently. On the other hand, using the polynomial hierarchy would have made the learning of the lower order models easier. From these results, we also see that structure learning accuracy gets worse with the average degree of the true model. This is behavior is consistent with the known lower bounds on the sample complexity of structure learning

These experiments also show that NeurISE has no problem finding the minima corresponding to the true model even when the number of samples is finite.

## C Results of learning the energy function with NeurISE.

In this section we will look at the results of learning the complete energy function using NeurISE by training it with the loss function given in Eq. (20). We will look at the results of using this loss on the Energy function in Eq. (16). Since we have the neural net representation for the full energy function we will compute the average loss in the energy function rather than in the conditionals. This comparison for a 10 variable model is given in Fig. 5a. We also compute the TVD between sampled distributions for the 15 variable model in Fig. 5b . The samples are now generated from the neural net using exact sampling rather than Gibbs sampling. This would have been intractable with neural nets approximations of the partial energy functions.

GRISE directly learns in the monomial basis, so the total energy function can be approximated by appropriately averaging the terms in the partial energy function. But this requires $p$ separate optimizations and increases the $N_p$ count of learning the energy function. To make the comparison with NeurISE more fair, instead we compute the $N_p$ value of $L$-order GRISE as $\sum_{k=1}^{L} \binom{p}{k}$. This is just the total number of independent parameters in a $L$-order energy function with $p$ variables.

From Fig. 5, we see that NeurISE learns the energy function well with less $N_p$. Notice that the neural nets used here are larger in size than that used in learning the partial energies. But here a single neural net learns the complete model, while in the other case we had $p$ separate nets learning the model.

(a)                                    (b)

Figure 5: Learning the full energy function of the model in Eq. (16) (a) Average $\ell_1$ error in energy for a 10 variable model (b) TVD between samples drawn from the learned models and those drawn from the true model for a 15 variable model

## Footnotes

[1]Code available at `https://github.com/lanl-ansi/NeurISE`.