[Reviews · NeurIPS 2020]

Review 1

Summary and Contributions: The paper alters the existing GRISE method for learning (discrete) graphical models to use NNs and empirically compares these two methods (and no others).

Strengths: There are some positive (empirical) results for NN-GRISE. It is not obvious that one can replace the basis function approach of GRISE with NN-GRISE and still recover graph structure; the authors show how to do it. The paper is competently written and effort has been made to allow reproducibility.

Weaknesses: The focus is narrow in that the goal is to extend GRISE to NN-GRISE and that's it. There is, for example, no comparison with any other approach to graph learning.

Correctness: Yes.

Clarity: Generally, however: When the authors use the term "graphical model" they actually mean "undirected graphical model". This should be made explicit early on in the paper. far less number of parameters -> far fewer parameters

Relation to Prior Work: Given the narrow focus of the paper it makes sense to mostly compare with earlier work on GRISE. There are some more general points about graph learning in the intro.

Reproducibility: Yes

Additional Feedback: Although the focus is narrow I do not see this as a big problem. The basic GRISE approach is to decompose an unsupervised learning problem into a collection of supervised learning problems. It is not obvious that a NN will be an OK choice for the supervised learning problems but the authors find a way and show that the flexibility of the NN (over a fixed collection of basis functions) has advantages. I think the authors should have made a bigger effort for "broad impact". Learning graphical structure can be scientifically useful but learning the wrong structure and trusting it can be a real problem. AFTER DISCUSSION/AUTHOR FEEDBACK Thanks to the authors for their feedback. It has not altered my mainly positive view of the paper.


Review 2

Summary and Contributions: This paper proposes a neural network-based algorithm to learn a discrete graphical model given i.i.d. samples. In particular, the proposed model NN-GRISE is an extension of the existing GRISE algorithm. The NN-GRISE takes advantage of the non-linear representation power of neural networks and represents the partial energy function with a neural net. By combining the neural network with GRISE, the proposed NN-GRISE method reduces the computational cost of GRISE, especially for models with high order interactions.

Strengths: Soundness of the claims: An intuitive justification for NN-ISO is provided which is helpful to understand the proposed model. Significance and novelty of the contribution: The proposed model applies a neural net to represent the potential function which is an interesting idea. In particular, the proposed model focuses on learning discrete graphical models. And with the neural networks, the computational cost compared to the original GRISE algorithm is reduced especially for models with high order interactions. The paper also shows that the proposed NN-GRISE can be applied to structure learning and for representing global energy function, which are both important aspects for learning discrete graphical models. Relevance to the NeurIPS community: The proposed model is relevant to the NeurIPS community.

Weaknesses: Soundness of the claims: 1. For parameter learning, the paper claims that NN-GRISE is able to learn the true partial energy functions given sufficient samples(line51). The paper also shows that NN-GRISE is a non-convex problem and true solution will be one of the local optimal solutions from NN-GRISE. One of the concerns is that there is no guarantee that the true solution can be obtained all the time. Without theoretical justifications and more experimental supports, the soundness of this claim is not clear. 2. For structure learning, the major concern is that the correspondence between the weights of neural networks and edges of graphical models is unclear. The paper provides only intuitive justifications, and experimental results are not strong enough to support the claim.  In addition, it is not clear how the threshold is chosen.  It would be better to give a more detailed theoretical explanation of how to relate the weights to the structures . And experiments on more datasets with different graphs should be provided to better support the claim. 3. The paper also claims that NN-GRISE can accurately reconstruct the neighborhoods of each variable for a general model with higher-order interactions(in the supplementary materials). There are no theoretical guarantees for this claim with limited experimental supports. 4. There is a lack of empirical evaluation. NN-GRISE is only compared against GRISE on synthetic data and small graphs. No comparison is provided between NN-GRISE and SoAs on benchmark real datasets. The experimental results are not representative. The authors should consider randomly generating structures of graphs, randomly generating the parameters to create synthetic data, repeating multiple times, and finally providing systematic results. Some parts of the experiment settings are not clear. For example, how the threshold for structure learning is decided? What is the value of the coefficient of the regularization term for structure learning? Significance and novelty: the novelty is limited. This paper is not the first work modeling conditional probabilities with NN. For example, “Masked gradient-based causal structure learning” and “Gradient-Based Neural Dag Learning” apply NN to learn conditional probability for continuous variables. The authors should discuss them as related works and consider including them into experiments for comparison to better demonstrate the effectiveness of the proposed model. Without further empirical demonstrations on more datasets compared to other SoA models, the significance of this work is not clear.

Correctness: Most of the claims and methods are intuitively ccorrect, as well as the empirical methodology. However, there are still some claims that need to be clarified. The empirical methodology also needs to be further demonstrated with more comprehensive experiments. The detailed comments are in the weaknesses section.

Clarity: This paper is overall well written.  There are small typos. In line 110, a period is missing after ‘the partial energy function’. The organization can be improved. It would be better to discuss proposed methods first including structure learning with NN-GRISE and total energy function representation and then show experimental results.  

Relation to Prior Work: It is clearly discussed how the proposed work differs from previous work (GRISE). The proposed method is an extension of the existing GRISE method. The GRISE algorithm is clearly introduced in section 2 with a discussion on its limitations. And the improvement of the proposed method is discussed with the empirical demonstration. However, we would suggest that the paper should have a discussion on other related works which also apply NN to graphical model learning, such as “Masked gradient-based causal structure learning” and “Gradient-Based Neural Dag Learning”.

Reproducibility: Yes

Additional Feedback: Detailed comments can be found in previous sections.


Review 3

Summary and Contributions: This paper introduces an approach of learning discrete graphical models (undirected) using neural networks. The idea extends an existing approach: Generalised Regularised Interaction Screening Estimators (GRISE) which is unsuitable for learning graphical models with higher-order interactions between random variables. This NN-GRISE algorithm is compared with GRISE. It is explained that a direct comparison is not trivial, yet, advantages and disadvantages are discussed.

Strengths: - Novelty of using a NN to represent the energy function of a graphical model. This may open up avenues for more research in this direction. - Clearly a relevant topic for NeurIPS

Weaknesses: The paper does not make it very clear what the significance is of the approach. This is also hard to assess given the current experiments, as there is only a fairly theoretical comparison with GRISE. Possibly, the authors could elucidate more on that in their rebuttal.

Correctness: As far as I can see, it is correct.

Clarity: The paper is fairly hard to read, and quite mathematical. However, I think it is acceptable.

Relation to Prior Work: This is discussed briefly in the introduction. However, I think given the claims on structure learning, a more extensive comparison (qualitative or experimental) is warranted.

Reproducibility: Yes

Additional Feedback:


Review 4

Summary and Contributions: This work builds on the past work called GRISE which learns graphical models with near optimal sample complexity by casting the learning problems as a convex optimization problem with constraints. GRISE although has near optimal sample complexity but suffers from very high computational complexity when higher order models are learnt even when there is a lot of symmetry present in the domain. The proposed work attempts to alleviate this key limitation by replacing the linear parameterization with a NN parameterization. The work shows learning a NN parameterization is highly effective in presence of symmetry in higher order models. The experiments are mostly proof of concept on a small number of variables in synthetic domains.

Strengths: The work presents a neural outlook and extension to the past work GRISE by learning the weights of basis functions by a neural network. This seems like a natural extension as many works in the past have similarly attempted to learn the factor weights by neural network. So, it seems the next logical step in this direction of learning graphical models. The experiments are although on synthetic domain but clearly illustrate the effectiveness of neural models wrt original GRISE. I like the additional experiments to visualize the structure as well and authors attempt to learn the complete model as well. So, the experiments on synthetic domains are complete and clearly illustrate the advantage neural models provide over the non-neural counterparts. The writing of the experimental section is clear and easy to follow. More on writing below here.

Weaknesses: I was not familiar with GRISE and my guess is that many members of the community will also not be familiar with this past work. So, I felt the background a bit unclear and difficult to follow. I will push authors to be a bit more precise and clear in notation even when they are reusing the notation. Also, it would be good to connect the notions defined with popular graphical models used extensively in literature. For e.g while defining the basis, it is not clear when authors say subscript s, does in mean subset of variable, a single variable of complete state of model? I presume, $\Theta$ could be shared between multiple variables in higher order models, so does optimizing it over one variable be sufficient without looking at other variables ? Although this question relates to GRISE , it would be great if authors can go in some detail for wider access to the community and keeping this manuscript complete. At many places, things are introduced Secondly, the experiments are rather weak with small numbers of variables like maximum 15 variables and only on synthetic data. Could these models be useful for some real world data and does learning higher models upto order 6 is required there ? I think neural models will require many more variables in the absence of this structure and it would be great to do some comparison on those lines as well. There is a lot of work on approximate lifted inference. Since experiments are done on highly structured data, some discussion of that related work which exploits this structure would be useful to connect to. “Lifted Inference and Learning in Statistical Relational Models” G Van den Broeck Three are no confidence intervals specified for neural experiments. How many seeds were tried and plotting confidence intervals could only tell much about statistical significance of results.

Correctness: Yes, the claims , method and empirical evaluation seem correct though no confidence intervals are plotted and only on synthetic data

Clarity: The paper is a bit hard to follow for a general reader not familiar with GRISE. Since the past work is a bit new, it would be ok to assume general readers to not assume familiarity with the past work and explain the past work more clearly grounding each notation. It would be great to add a conclusion section by compressing the experiments a bit. The paper seems to end abruptly in the current version.

Relation to Prior Work: Related work relates with only past work on GRISE etc is discussed. If the paper’s main assumption is in assuming some structure in the underlying model, it should expand on other related works in graphical models and statistical relational learning.

Reproducibility: Yes

Additional Feedback: Although code is provided, it would be great to include all details like how many seeds used etc. in the text as well. -------------------------------------- Thanks for the rebuttal. The rebuttal addressed most of my concerns.

[Author Response · NeurIPS 2020]

We thank the reviewers for their helpful comments and suggestions. Our responses below address the main suggestions for improvement proposed by the reviewers, in particular: clarification of claims, choice of the baseline to compare against (GRISE), and additional experiments that show that our methods work for randomly-generated instances.

**Reviewer #1:**
**Comparison with other methods**: We chose to use comparison with GRISE only as a baseline because this algorithm is currently the state-of-the-art for learning for learning undirected graphical models (both theoretically from the sample-complexity point of view [19] and empirically [11]), beating other approaches. Also, while we show in this paper that a particular form of GRISE makes it possible to generalize it to include NN representation of the energy function, it is currently unknown whether for other methods this can be done. This broadens the scope of our contribution. **Broader impact**: We agree with reviewer's suggestion on the broader statement, we will improve it for the camera ready version.

**Reviewer # 2:**
**Non-convexity:** Although NN-GRISE is non-convex, we show that the true solution will be at the *global optima* of the loss function if the neural net can represent the true model and if there are enough samples [lines 140-142]. This ensures that the true solution can be obtained with stochastic gradient descent type methods if the neural net is large enough, as shown for many classes of neural network problems at a series of works at previous NeurIPS conferences. We agree that this is an important point and we will highlight it in the final version. We find from our experiments that size of neural nets needed to get to this limit is favorable compared to expanding in the monomial basis for models with higher order interactions and symmetries. **Structure identification claims:** Let us clarify the justification of our claim: We show that if we work with a representative enough neural net, then global optimum is guaranteed to encode the neighborhood information of the model, *even for higher order models*. From the conditional independence property, the output of the NN should not depend on the non-neighbors, and the input regularization biases the training towards the minimum with this structure. This principle is clearly demonstrated in the experiments. Following the reviewer's suggestion, we provide a summary figure with additional experiments for structure learning for pairwise and higher-order models (Fig. 1), where for each point we consider 20 randomized problems with more variables compared to the manuscript.

Figure 1: Accuracy of structure learning for models with pairwise couplings, 30 variables (Left), and higher-order interactions, 20 variables (model described in the supplementary section) (Right). "Total accuracy" accounts for both edges and non-edges.

**Additional empirical evaluation:** Our goal in the paper was to show representative examples that illustrate each of the claims on a concrete example. We agree that providing evidence that the algorithm works for general graphs will strengthen the presentation, which we do with additional experimental results (Fig. 1). For the regularization parameter we used the one for the GRISE algorithm [18,19], as the constant's scaling comes with strong theoretical arguments. Selection of the threshold in practice can be a delicate problem related to the gap to the weakest coupling in the graph, see [11] for a thorough discussion of the setup. Importantly, experiments in this paper show an emergence of such a gap. The threshold for detecting the edges can be chosen either by inspecting the histogram of trained weights or using a standard-deviation based outlier detection method (used in the experiments above). **Discussion of related work:** Our work focuses on undirected graphical models, and not DAGs. Instead of using NNs to directly learn the conditional probabilities, our method uncovers parsimonious basis representations for an unknown class of models. We will make sure to discuss related approaches for continuous variables and their differences with ours in the revised version.

**Reviewer #3: Additional experiments:** We focus on providing intuitive examples illustrating each of the key points. We agree that a stronger experimental evidence will improve the perception of the narrative. Following the reviewer's recommendation, we provide additional results over the ensemble of networks (see Fig. 1).

**Reviewer #4: Presentation:** We agree that the presentation of GRISE and problem setting could be done even more accessible for a general reader. We will work on this and discuss lifted inference (including the reference suggested by the reviewer) in the camera-ready version. **Additional experiments:** Applications involving multi-body interactions are numerous, e.g. in natural systems; we use an example of a real problem in quantum computing for an illustration, with a primary goal to showcase our method in a controlled synthetic setting with ground truth. Following the reviewer's recommendation, we produced additional aggregated plots that include confidence intervals and show that our method works on randomized graph instances (see Fig. 1). In the camera-ready version of the paper, we will also clarify that no seed use is needed, all results are obtained through a single run of our algorithm with zero couplings initialization.

[Meta-Review · NeurIPS 2020]

The discussion has converged to a positive evaluation by the reviewers, in particular because the authors' response clarified some important points. In view of that, the authors are strongly invited to take the feedback on board for the final version. There have been some clear points for improvement (position wrt SOTA - comparing only to GRISE has been a point, guarantees or lack of them, attempts to discuss the generality and broader impact beyond learning a structure, etc).